# Reclaiming Indigenous Health in the US: Moving beyond the Social Determinants of Health

**DOI:** 10.3390/ijerph19127495

**Published:** 2022-06-18

**Authors:** Stephanie Russo Carroll, Michele Suina, Mary Beth Jäger, Jessica Black, Stephen Cornell, Angela A. Gonzales, Miriam Jorgensen, Nancy Lynn Palmanteer-Holder, Jennifer S. De La Rosa, Nicolette I. Teufel-Shone

**Affiliations:** 1Native Nations Institute, Udall Center for Studies in Public Policy, University of Arizona, Tucson, AZ 85719, USA; stephaniecarroll@arizona.edu (S.R.C.); scornell@arizona.edu (S.C.); mjorgens@arizona.edu (M.J.); 2Mel & Enid Zuckerman College of Public Health, University of Arizona, Tucson, AZ 85724, USA; 3Good Health Wellness in Indian Country Program, Albuquerque Area Southwest Tribal Epidemiology Center, Albuquerque, NM 87110, USA; msuina@aaihb.org; 4Department of Alaska Native Studies and Rural Development and Tribal Governance, University of Alaska, Fairbanks, AK 99775, USA; jcblack@alaska.edu; 5Department of Sociology, University of Arizona, Tucson, AZ 85719, USA; 6Udall Center for Studies in Public Policy, University of Arizona, Tucson, AZ 85719, USA; 7School of Social Transformation, Arizona State University, Tempe, AZ 85281, USA; aagonz31@asu.edu; 8BALCSTAR Consulting PLLC, Omak, WA 98841, USA; balcstar.llc@hotmail.com; 9University of Washington, Seattle, WA 98195, USA; 10Family and Community Medicine, College of Medicine Tucson, University of Arizona, Tucson, AZ 85711, USA; jschult1@arizona.edu; 11Comprehensive Pain & Addiction Center, University of Arizona, Tucson, AZ 85724, USA; 12Sonoran Center for Excellence in Disabilities, College of Medicine Tucson, University of Arizona, Tucson, AZ 85719, USA; 13Center for Health Equity Research, Northern Arizona University, Flagstaff, AZ 86011, USA; nicky.teufel@nau.edu; 14Department of Health Sciences, Northern Arizona University, Flagstaff, AZ 86011, USA

**Keywords:** social determinants of health, Indigenous knowledge, WHO, healthy community, health, Indigenous

## Abstract

The lack of literature on Indigenous conceptions of health and the social determinants of health (SDH) for US Indigenous communities limits available information for Indigenous nations as they set policy and allocate resources to improve the health of their citizens. In 2015, eight scholars from tribal communities and mainstream educational institutions convened to examine: the limitations of applying the World Health Organization’s (WHO) SDH framework in Indigenous communities; Indigenizing the WHO SDH framework; and Indigenous conceptions of *a healthy community*. Participants critiqued the assumptions within the WHO SDH framework that did not cohere with Indigenous knowledges and epistemologies and created a schematic for conceptualizing health and categorizing its determinants. As Indigenous nations pursue a policy role in health and seek to improve the health and wellness of their nations’ citizens, definitions of Indigenous health and well-being should be community-driven and Indigenous-nation based. Policies and practices for Indigenous nations and Indigenous communities should reflect and arise from sovereignty and a comprehensive understanding of the nations and communities’ conceptions of health and its determinants beyond the SDH.

## 1. Introduction

Indigenous communities support healthy, vibrant lives embedded in their own Indigenous knowledge, values, and traditions. Even today, despite settler-colonial efforts to either wipe out or totally assimilate individuals and collectives, Indigenous nations continue to practice and revitalize their traditional knowledge and values to bring health and well-being to their communities and convey knowledge to future generations. However, health inequalities between Indigenous and the dominant, power-holding, settler, white populations have existed for decades in the United States [1,2]. Indigenous morbidity and mortality are concerns not only for US public health policy but especially for US Indigenous nations themselves as they set policies and practices to protect and promote the health and well-being of their citizens [3,4]. Tribal government efforts to address inequalities in morbidity and mortality have included investing in and developing tribal health departments, managing health care services, and engaging with health research activities to protect against ethical violations, allow greater community control of resources, data, and methods, and encourage research to address tribal concerns [5,6,7,8]. However, Indigenous nation health departments, control of health care services, and stewardship of research may not be sufficient to effectively address tribal health concerns. Tribal actions focused solely on increased control of health-related services and research do not adequately address either the underlying determinants of health or the question of what, in Indigenous conceptions, constitutes a healthy society [4,9,10,11,12]. 

In February of 2015, the Native Nations Institute (NNI) convened scholars (referred to as the panel throughout this paper) on the social determinants of health (SDH) to consider these issues. The NNI, a research and outreach unit at the University of Arizona, strengthens Indigenous governance through research and policy analysis, educational programs, and tribal services. Scholars included Indigenous and non-Indigenous tribal health professionals and academic researchers. The goal of the convening was to explore Indigenous conceptions of health and an Indigenous health determinants framework. This paper presents the outcomes of those discussions, including an examination of the assumptions within the World Health Organization’s (WHO) SDH framework, a call to refocus on Indigenous conceptions of healthy communities, and the need to identify Indigenous determinants of health that center on sovereignty, Indigenous ways of knowing, and utilizing Western knowledge, as needed. The age of data is a possible limitation of the paper’s discussion of these outcomes.

The positionality of individuals on the panel influenced the construction of the thoughts in this paper [13,14]. Diverse in culture, academic discipline, and life experiences, the eight scholars who participated in the panel represented five US Indigenous nations, Ahtna—Native Village of Kluti-Kaah, Cochiti Pueblo, Gwichyaa Zee Gwich’in—Native Village of Fort-Yukon, Hopi, and Colville; three allied settler individuals; and the academic disciplines of education, justice and social inquiry, political economics, public health, social work, and sociology. The ideas in this paper emerged from the convergence of each participant’s contribution based on their position in the world and the group [13,14,15]. All eight panelists authored this paper, along with two authors who joined to assist in the writing process. One of those authors represents the Citizen Potawatomi Nation, while the other is an allied settler individual. The results reflect different perspectives formed into shared ideas and epistemological pluralism of disciplines and cultures [13,14,15].

## 2. Indigenous Nations and Social Determinants of Health

The WHO has stated that the social conditions in which individuals grow, live, and age often have a greater impact on health than behaviors, genetics, or health care [16]. Early childhood experiences, social inequality and social exclusion, security of access to food and water, stress, and the availability of and access to employment are among the social characteristics that have been shown to affect health outcomes for individuals and communities worldwide [17,18,19]. This is also the case for Indigenous communities. For example, in Australia; Canada; and Aotearoa New Zealand, growing attention is being paid to the social determinants for Indigenous Peoples’ health statuses and outcomes [20,21,22,23,24,25,26,27,28,29,30,31,32,33]. Indigenous communities’ unique historic, social, and political experiences yield distinctive social determinants of health such as self-determination; settler colonialism; migration; globalization; cultural continuity and attachment; relationships with land and non-human relatives; social support, capital, and cohesion; racism and social exclusion; and justice systems [9,10,11,12,34,35,36,37,38,39,40,41,42,43,44].

However, these topics have received less attention in the US, where the majority of health research dollars are spent developing medical technology and assessing the impact of health services and interventions among mainstream populations [45,46]. While some tribal leaders in the US have urged that Indigenous health research and policy pay more attention to the social determinants of health, published explorations of the topic, and in particular of their effects on Indigenous health, are rare [4].

Furthermore, it is not clear that the WHO SDH framework adequately captures Indigenous Peoples’ lifeways. Many Indigenous communities either have experienced or continue to experience other factors that can affect health outcomes, but that is dealt with only obliquely or not at all in the WHO SDH framework. Among those particular to Indigenous communities are radical disruptions of traditional relationships and cultural practices; loss of autonomy and subjection to intrusive external administration of community life, such as lack of governance over land in Alaska; boarding schools and forced migration that cause separation from culturally, socially, and economically significant lands and community; physical violence; and racism, and the often lasting trauma associated with these experiences [9,10,11,12,26,41,42,47]. In addition, Canadian researchers found that the SDH framework fails to account for the full effects of settler colonialism on the health of Indigenous Peoples and that the field of SHD lacks sustained inquiry into the determinants of health specific to Indigenous Peoples, particularly research and writing by Indigenous people [47].

Such arguments have led the US and international researchers to call for more information on the particular determinants of health that affect Indigenous communities [9,10,11,12,34,35,36,37,38,41,48].

## 3. What Is Health?

Since 1947, the WHO has promoted a comprehensive view of health that includes an individual’s social and mental well-being in addition to physical health and the absence of disease [49]. As the understanding of health status and outcomes evolved, so did the understanding of how society influences the health of an individual. By the turn of the Century, the SDH framework dominated how social science and medicine intervened to improve health. As applied, the framework conceives of health as an equation or a system with inputs and outputs: the determinants are on the left side of the causal arrow with society’s health status or outcomes, primarily indicated by aggregated measures of individuals, on the right side [50]. While turning to the SDH, the general concepts of what constitutes health remained stagnant and continued to align with the 1947 definition: focused on the individual and including only mental, physical, and social health.

In Indigenous societies, discussions in the late Twentieth Century began to expand ideas about healthy individuals and societies beyond the SDH to include other determinants as well as the health of a collective (e.g., family, community). In Australia, the National Aboriginal Health Strategy set in 1989 further Indigenized the WHO approach by adding cultural well-being and community health to the definition of Aboriginal health [24,29]. In an updated version adopted by the National Aboriginal Community Controlled Health Organization Constitution as amended in 2006, “‘Aboriginal health’ means not just the physical well-being of an individual but refers to the social, emotional and cultural well-being of the whole Community in which each individual is able to achieve their full potential as a human being thereby bringing about the total well-being of their Community. This holistic approach includes the cyclical concept of life-death-life” [23] (pp. 5–6). Comparable efforts to rethink the nature of a healthy society are also apparent in Canada and Aotearoa New Zealand. Similarities in these movements include Indigenous actions embracing non-linear relationships between health and illnesses, a focus on collective health, and holistic solutions to improving community health and well-being that utilize efforts beyond health care and public health [20,22,23,24,25,27,28,29,34,47].

These efforts to reframe health in Indigenous terms embrace holistic and ecological aspects of many Indigenous knowledge systems, including the recognition that individuals live within a web of relationships that impact well-being. These relationships include not only other people and the community at large but the natural and spiritual worlds, as well as past events and experiences that have long-term, health-related impacts on families and communities [26,32]. A healthy society, in this view, is not simply one in which individuals are free of disease, but one in which relationships are functioning in healthy and productive ways. Furthermore, an effort to free individuals of disease that ignores the accumulated knowledge of such relationships and how they work may have only limited success [20,22,23,24,25,27,28,29,31]. These ideas have surfaced in the discussion of Indigenous health frameworks in the US. For example, the National Institutes of Health and some tribes have embraced the Medicine Wheel as a pan-Indigenous or nation-based approach to holistic health that includes physical, mental, emotional, and spiritual dimensions [51,52]. However, the Medicine Wheel does not encompass the variety of North American Indigenous knowledge systems, which seldom appear in the peer-reviewed US literature on Indigenous health and well-being [53]. 

## 4. Methods

The research method consisted of a consensus panel conducted via standard methods. Convened in Tucson, AZ, 17 and 18 February 2015, the consensus panel narrowly focused on creating normative feedback among experts on the social determinants of health in US Indigenous nations about the limitations of the social determinant of health framework for Indigenous communities, Indigenizing the WHO social determinants of health framework [18,54], and Indigenous conceptions of *a healthy community* [55,56,57].

The consensus panel comprised project researchers and invited experts as co-learners and co-producers of ideas. Invitees included select scholars engaged in the topics of Indigenous health and social determinants identified via researcher networks. The research project paid participant travel costs and provided a stipend. Prior to the meeting, the lead researcher created and provided participants with a meeting agenda, a list of participants, and a draft manuscript outline of the background literature on Indigenous health and social determinants.

The researchers assured reliability and validity of the methods and results through an iterative process affecting all aspects of the consensus panel, including a review of the relevant literature and other documents and discussions with other researchers and tribal leaders, program directors, and staff. The process included creating the agenda and manuscript outline based on the available literature. The agenda and manuscript outline acted as the consensus panel guide. Standard note-taking procedures were followed during the panel, which researchers used to produce a written summary document. Analysis of the topics and consensus occurred through an iterative process of discussion during and after the panel. The researcher verified the information via conversations over email, phone, or videoconferencing. This method of assuring reliability and validity resonates with both Western qualitative research methods and Indigenous methodologies [13,55,56,57,58,59,60].

## 5. Rethinking the Social Determinants of Health Framework

Efforts to Indigenize the WHO SDH framework do not interrogate assumptions that undergird the model. Table 1 identifies assumptions as reported in the literature and as recognized by the panel participants [18,19]. Generally, the WHO SDH framework, even when adapted to Indigenous circumstances, continues to use Western ways of knowing and values to define determinants, health, and well-being and describe the relationships among these terms. In addition, the panel convened by NNI felt that the WHO SDH framework used an active voice of *the other—the dominant, white, settler population*—as those working, helping, and saving to reach health equity for a subpopulation, e.g., Indigenous Peoples, instead of leading with communities’ knowledge and episteme [61].

Panel participants identified underlying Western concepts in the WHO SDH framework as descriptive, prescriptive, and linear resulting in a model that perpetuates discipline and system segregation, defines determinants to operationalize, and suggests that improvements in determinants lead to enhanced community health. In contrast, Indigenous knowledges argue for an action-oriented process that uncovers the holistic network of interconnected determinants of health and well-being for Indigenous nations. This process would look to improve Indigenous nations’ policies via sovereign, self-determined, community-based actions to strengthen culture, traditions, languages, and social ties.

Table 1 relays how the social determinants of the health agenda pertain to the WHO “closing the gap” strategy that informs policy actions taken to reduce inequalities between subpopulations (e.g., all Indigenous Peoples) and the mainstream or dominant population [62]. Within the “closing the gap” paradigm, focusing on describing Indigenous communities via their own data or comparisons between Indigenous populations rarely occurs in the US or internationally [9,11,62]. Describing Indigenous communities through their own lens and via their own data encourages focusing on protective factors within a community that has led to long-term resilience, such as ceremonies, traditional roles and responsibilities, and rites of passage. Comparing Indigenous nations *on their own terms* could reveal key successes or challenges unique to Native communities and nations, likely as a result of settler colonialism, such as increasing language fluency to improve consumption of traditional foods for better diabetic management. Another limitation of “closing the gap” is that using only population-based data comprised of aggregated individual-level measurements to compare Indigenous populations with the US white population does not allow Indigenous nations to conceptualize appropriate metrics for determinants, health, and well-being that resonate with the community, reflect culture and traditions, and provide meaningful insight into local experiences. Examples of these metrics include measures of non-human health for land and animals, data on spiritual and cultural health such as language and sacred sites, and indices for determinants that reflect Indigenous realities such as a collective orientation and community conceptions of wealth and jobs. Notably, over the past decade, traction to Indigenize the “closing the gap” strategy has begun to disrupt the WHO paradigm. In Australia, the Close the Gap strategy was completely reviewed and reshaped to provide greater power and agency to Aboriginal communities and Aboriginal Community Controlled Health Organisations [63]. Moreover, the new Australian National Aboriginal health plan was launched, which situates culture and community at the center [64].

Critiques of employing the WHO SDH framework in Indigenous communities noted that the disconnect between Western and Indigenous knowledges and epistemologies results in the inability to capture holistic and land-based Indigenous health beliefs [9,11]. The WHO SDH conversation tends to begin and end with measurable physical health outcomes such as morbidity and mortality; in short, on deficits [61]. Panel participants underscored the importance of recognizing that health and well-being within Indigenous nations may be bolstered by their considerable assets, such as ties to the land, intergenerational transfer of knowledge, traditional foods and medicines, and ceremonies; asset-based and non-physical measures of health may be appropriate for Indigenous communities. In fact, leveraging assets and protective factors may provide better traction in addressing community health issues than deterring risks [43]. Progress in this realm has occurred as part of informing the United Nations Sustainable Development Goals (SDG). For example, reports criticized the SDG effort for not including Indigenous conceptions of well-being and only mentioning Indigenous Peoples in 4 of 169 metrics [65,66]. While calls were made for the broader inclusion of Indigenous Peoples in the SDG process, Indigenous communities are charting their own paths by creating and using their own metrics, and their contributions measurements for the SDG were realized [65,66]. 

Panel participants also noted that the WHO SDH framework tends to focus on the individual and, for the most part, uses aggregate individual measures to approximate collective community health and well-being [9,11,67]. It is on the community that analysis should center [9,11]. Aggregate individual health outcomes, e.g., disease rates, do not reflect collective health outcomes. Collective outcomes might include the existence of trash in the natural landscape, language classes offered at local schools, or the offering of traditional foods at community events.

Relatedly, the WHO framework is removed or decontextualized from time and spiritual space and lacks certain relational considerations [9,11]. Indigenous knowledge incorporates intergenerational roles and responsibilities into the community’s vision, history, and spiritual space, including relations with and sacred responsibilities to ancestors and those yet born as well as the land [9,11,26,32,68]. For example, Indigenous scholar Vine Deloria writes about the reluctance to surrender lands, in part, because of Indigenous Peoples’ connection not only to the land but also to their ancestors who still spiritually inhabited the land [69]. 

The panel noted that the WHO SDH framework assumes that determinants and health outcomes data are available for analysis. Indigenous nations face data challenges when attempting to measure both mainstream and Indigenous-specific health determinants and health outcomes. Virtually all published efforts focused on rethinking the social determinants of health for Indigenous peoples discuss the poor quality of existing data and limited Indigenous nation-level data [9,11,34,35,36,37,38,48]. The panel expressed that in addition to a dearth of information for Indigenous nation-level decision-making, the lack of data hinders the development of a specifically Indigenous framework and limits comparative research. The data that are available do not usually address nation or Indigenous-specific determinants of health or visions of a healthy society. The panel underscored the importance of Indigenous nations exercising Indigenous data sovereignty and data governance, in part through financial investments and collaborations with federal, state, and other governments as well as non-profits to invest in community-based data collection, analysis, and use data that address tribal needs and aspirations.

The panel observed that one model does not work for all communities. The mainstream WHO SDH framework too often lacks local specific context. Indigenous nations are diverse in population size, land base, history, location, and political, social, and cultural structures. This diversity requires numerous disparate policies and Indigenous nation-based actions. The panel concluded that a holistic framework focused on community conceptions of health and well-being would allow for including the appropriate array of health and wellness determinants for each Indigenous nation.

## 6. Conceptualizing Indigenous Health and Health Determinants

The panel decided that simply modifying the framework employing Indigenous knowledge(s) was not adequate to inform Indigenous nation policy and action to enhance, sustain, and support health and wellness in Indigenous communities. Panel participants felt that any discussion of health determinants would need to take into account Indigenous-specific conceptions of a healthy society. While the panel was reluctant to propose an Indigenous-specific framework, the participants underscored the use of holistic perspectives that categorize health and well-being determinants. The underlying knowledge and epistemologies should reflect the community’s ways of knowing—be community-driven—and the nation’s inherent rights to self-determine strategies and mechanisms to address health—be nation-based and sovereign. As such, conceptions of health and well-being could then inform how health, well-being, and determinants are measured, assessed, and compared for that nation and its communities, across Native nations, and in comparison to mainstream metrics and measurements.

In this perspective, conceptions of health and well-being establish the determinants of health, which include social as well as other community-defined determinants. The panel categorized determinants of health and well-being into three categories: (1) *broad determinants* of health that affect Indigenous and non-Indigenous communities; (2) *shared determinants* of health among Indigenous communities or among communities in a certain geography or of a certain culture, e.g., US Midwest communities; and (3) *unique determinants* of health evident in one or a few Indigenous or other communities. The schematic in Figure 1 provides space for a variety of determinants *and* a holistic concept of health that may include social, cultural, emotional, and other types of health as determined by the community.

Unique determinants are particular to each Indigenous nation, such as culture, use of natural resources for health and healing, traditional practices and ceremonies, and language. Shared determinants include resilience; relationships with ancestors and future generations; the interconnectedness of determinants and health and well-being; relationality; an orientation toward the collective; the individual’s role in the collective; interdependence; the importance of and relationships with ancestral and other lands, place, and space; the significance of elders; the intergenerational transmission of traditional knowledge; the changing meaning of health over the life course; nation self-determination and sovereignty; and colonization [9,10,11,12,34,35,36,37,38,41,70].

Broad determinants include those identified by the WHO and others as playing a role in creating the environments where people grow, live, and age, such as governance, community cohesion, jobs, health, and other services [16,18].

Figure 1 also elucidates the relationships between the three categories and health and well-being viewed holistically. Unique and shared determinants may overlap. A common environmental context may influence health and healing in many nations that exist in a shared ecosystem, e.g., sage grows in the desert, and many but not all Indigenous nations use sage. Cultures or languages are shared among a few nations, but not all. Shared and broad determinants also overlap. Mainstream broad determinants, e.g., governance, community cohesion, jobs, health, and other services, while applicable, may be conceptualized and measured differently for Indigenous communities. The framework in Figure 1 is not prescriptive. However, it represents a process that establishes community-driven conceptions of health and well-being. As it begins with locally defined health and well-being, the framework supports a course of action that incorporates partnerships and collaborations to identify unique, nation-based determinants; defines shared determinants that drive comparison and innovation among Indigenous nations; and employs or adjusts broad determinants to allow comparisons across Indigenous nations or with other US populations. The schematic in Figure 1 utilizes Indigenous knowledges, cultures, and epistemologies to co-create with Western ideas, community-driven, nation-based theories of health and its determinants.

## 7. Discussion

Moving towards an Indigenous framework for understanding the determinants of health and well-being in Indigenous communities requires action from and presents challenges for Indigenous nations and federal, state, and private policymakers, funders, and collaborators. Reclaiming Indigenous health begins with community-driven, nation-based processes grounded in sovereignty and self-determination. For Indigenous nations to create and sustain healthy societies, nations must comprehensively assess collective conceptions of health grounded in community knowledge and epistemologies that access other ways of knowing, such as Western science, toward community goals and visions. Such grounding in community knowledge of health and well-being allows Indigenous nations to identify unique, shared, and broad health indicators and determinants. In addition to nation-based action, collaboration among Indigenous nations must occur in order to create metrics and measurements for shared determinants that enable comparisons across nations on Indigenous Peoples’ own terms.

Nation-based efforts do not require that Indigenous nations enact every policy or fund every program; sovereignty and self-determination assure that Indigenous nations have the inherent right to make the decisions and take action to set priorities, enact policy, seek outside funds, spend funds, and partner with other entities to meet the nation’s goals.

Challenges for Indigenous nations include securing funding for community planning and development; developing policies and designing programs and practices to align with community-driven, nation-based priorities; communicating Indigenous conceptions of health and well-being to partners, collaborators, and funders; and considering how community-driven, nation-based processes and conceptions apply to Indigenous nation non-resident citizens. Along with these challenges come opportunities for involving community members via nation-specific, culturally appropriate methods and practices in discussion and collective decision-making about what it means to be healthy in a holistic, community-oriented context.

Federal, state, local, and other governments, as well as non-profits and other funders and partner entities, should support Indigenous nation sovereignty and self-determination regarding the health and well-being of Indigenous communities. This support calls for reframing and allowing for differences in how health is conceptualized in policies, reports, requests for proposals, funding, research, programs, partnerships, and relationships with Indigenous nations and peoples. In addition, a shift from funding primary health services and interventions to more flexible financial support for planning and systems improvements requires infrastructure investments. Challenges for the federal government and other funders, collaborators, and partners include flexibility given the plethora of differences between Indigenous nations and the need for Indigenous nations to determine actions on nation-specific determinants, and establishing and maintaining partnerships based on self-determined determinants of community health and well-being.

## 8. Conclusions

The differences between Indigenous knowledge concerning health and well-being and Western ways of knowing that underpin the WHO SDH framework argue for an iterative process to incorporate Indigenous knowledge and practices concerning health and well-being. Comprehensive community-driven, nation-based reclaiming and defining of Indigenous health and well-being is necessary to establish and address the broad array of determinants of health and well-being in Indigenous communities. An Indigenous framework will support capable governance and inform policymaking at tribal, federal, and other levels to realize healthy Indigenous communities. At their core, these frameworks and the policies recognize that protecting and promoting Indigenous Peoples’ health demands more than equity and closing the gaps. Sovereign, self-determined actions by Indigenous nations and communities form the foundation for sustainable collective well-being.

## Figures and Tables

**Figure 1 ijerph-19-07495-f001:**
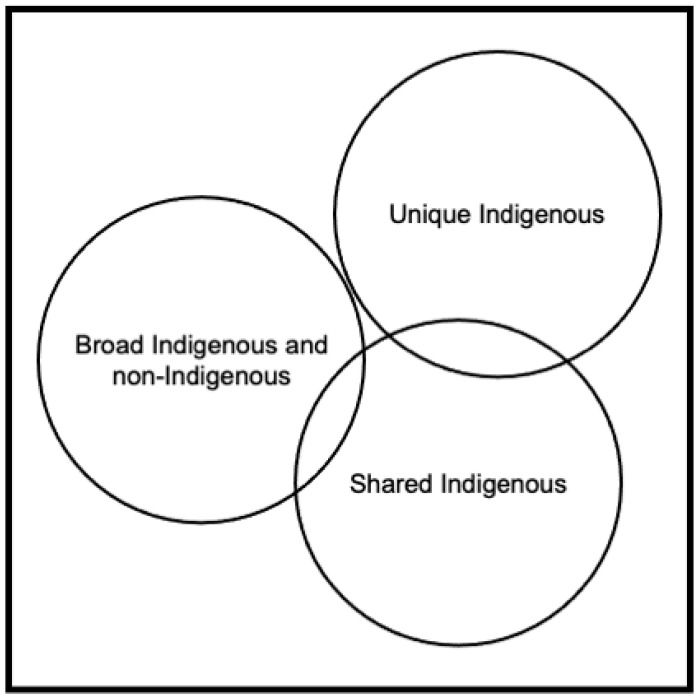
Determinants of Collective Health and Well-being in US Indigenous Communities.

**Table 1 ijerph-19-07495-t001:** Determinants of Health: Indigenous Knowledge and Western World Views.

Non-Indigenous Knowledge Base in the WHO Social Determinants of Health Framework	Indigenous Knowledge of Determinants, Health, and Well-Being
Based on Western values of framework	Connects with community values, language, culture, land, place, stewardship
Voice of the “other”	Indigenous Voice
Descriptive	Action oriented
Prescriptive	Community determined
Linear	Holistic
Focuses on “closing the gaps” between subpopulations and the general or dominant population	Aligns movement with the community’s own vision of a healthy, sustainable society
Broadly applicable to all communities	Flexible for application in many communities
Decontextualized in time and spiritual space	Incorporates history and spiritual place
Lacks relational considerations among people and between people and non-human world	Considers future generations and ancestors, intergenerational, including a role for each community member
Distinctions made between social, individual, biological and genetic, physical and other determinants	Interconnectedness of all determinants
Focused on the individual	Focused on the collective, and the individual’s role in the collective
Determinant indicators and health outcomes primarily Western-defined disease prevalence and incidence rates, economics, education, and other measures	Metrics and measurements reflect Indigenous conceptions of health and society, including Indigenous community-specific economic activities, e.g., individual or small business art production and sales; tourism
Determinants indicators and health outcomes assume that communities have access to data to measure, assess, and track progress	Lack of data available at the nation, reservations, and tribal citizen levels; Indigenous data need to reflect Indigenous conceptions of health, well-being, and determinants
Deficit based	Asset based
Disease based	Health based

## Data Availability

Not applicable.

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
