# Peer review of "Reclaiming Indigenous Health in the US: Moving beyond the Social Determinants of Health"

_ijerph, 2022, doi:10.3390/ijerph19127495_

Round 1

Reviewer 1 Report

The authors use the term "white populations" (lines 48;173;198) which is very generalized. As much as it is known in the history of USA and respectfully, the term should be changed to "dominant population; majority population". Facts: The article presents the Present and not the Past. Nowadays, the USA population (2022) is very diverse including peoples of other races, cultures and religions who also disregard the culture, health and interests of the indigenous populations. If the goal of the manuscript is:  "Moving Beyond the Social Determinants of Health" as mentioned in the Title and throughout the article, a more inclusive, actual highly diverse population, and less separatist will bring a positive transformation and change in the society to achieve the very important health equality and equity in the USA to all peoples. 

Author Response

Reviewer One

The authors use the term "white populations" (lines 48;173;198) which is very generalized. As much as it is known in the history of USA and respectfully, the term should be changed to "dominant population; majority population". Facts: The article presents the Present and not the Past. Nowadays, the USA population (2022) is very diverse including peoples of other races, cultures and religions who also disregard the culture, health and interests of the indigenous populations. If the goal of the manuscript is:  "Moving Beyond the Social Determinants of Health" as mentioned in the Title and throughout the article, a more inclusive, actual highly diverse population, and less separatist will bring a positive transformation and change in the society to achieve the very important health equality and equity in the USA to all peoples. 

Response to Reviewer One

Sometimes we are comparing to the dominant structures, which were created by white populations so that is why white is used. In other places the term settler was used instead.

Yes, the USA population is diverse. The first part of the manuscript title is Reclaiming Indigenous Health in the US, which indicates that the manuscript is focused on Indigenous populations in the US these populations health equality and equity. By raising the health equality and equity of the Indigenous populations will also up the US overall achieve important health equality and equity.

Reviewer 2 Report

Thank you for the opportunity to read this very interesting and thoughtful paper. It is critically important that WHO frameworks are reimagined and redeveloped through contemporary culturally led processes that challenge the 'one size fits all' thinking for healthcare design and assessment.

I have some minor and major reflections for consideration.

Major

  1. I am concerned at the age of the data (2014). So much has changed in Indigenous health research and methodologies since then. For example, the Close the Gap strategy in Australia has been completely reviewed and reshaped to provide greater power and agency to Aboriginal communities and Aboriginal Community Controlled Health Organisations. In parallel and new Australian National Aboriginal health plan has been launched which situates culture and community at the centre.  It provides a wonderful model for re-thinking healthcare National Aboriginal and Torres Strait Islander Health Plan 2021–2031 | Australian Government Department of Health that may be of value to the author team in disrupting WHO frameworks.  

    I suspect there are major policy and strategy changes in other high income colonised countries that would also be relevant to the work here.

  2. Figure 1 is not clear - it is unclear how the words that sit outside the Venn diagram relate to the circles and how does this relate to conceptualisation of health and healthcare.  I urge the author team to consider their figure in the context of the example provided above and other similar work. 
  3. Methods require further detail - what was the process end to end including recruitment, data collection and analysis and positionality of investigators/participants (I really like your positionality statement line 74).  

Minor

  1. Line 48 Indigenous and non-Indigenous, white - I am not clear as to why the authors have added 'white' as I suspect the data in high income colonised countries, Asian citizens for example would have similar health outcomes to white citizens. I appreciate that this may be different in the US and suggest that a note to explain this is warranted.

  2. Line 215 - says 'risky', I suspect you mean 'risk'. 

I note that I am an ally who lives in a different country and that I have no experience in the US.

I think this paper is an important one but it needs a lot more work to reflect and represent  your data in the content of contemporary evidence and frameworks for (re)conceptualising healthcare policy and practice through a cultural lens. 

Author Response

Reviewer Two

  1. I am concerned at the age of the data (2014). So much has changed in Indigenous health research and methodologies since then. For example, the Close the Gap strategy in Australia has been completely reviewed and reshaped to provide greater power and agency to Aboriginal communities and Aboriginal Community Controlled Health Organisations. In parallel and new Australian National Aboriginal health plan has been launched which situates culture and community at the centre.  It provides a wonderful model for re-thinking healthcare National Aboriginal and Torres Strait Islander Health Plan 2021–2031 | Australian Government Department of Health that may be of value to the author team in disrupting WHO frameworks.  

    I suspect there are major policy and strategy changes in other high income colonised countries that would also be relevant to the work here.

  2. Figure 1 is not clear - it is unclear how the words that sit outside the Venn diagram relate to the circles and how does this relate to conceptualisation of health and healthcare.  I urge the author team to consider their figure in the context of the example provided above and other similar work. 
  3. Methods require further detail - what was the process end to end including recruitment, data collection and analysis and positionality of investigators/participants (I really like your positionality statement line 74).  

Minor

  1. Line 48 Indigenous and non-Indigenous, white - I am not clear as to why the authors have added 'white' as I suspect the data in high income colonised countries, Asian citizens for example would have similar health outcomes to white citizens. I appreciate that this may be different in the US and suggest that a note to explain this is warranted.
  2. Line 215 - says 'risky', I suspect you mean 'risk'. 

I note that I am an ally who lives in a different country and that I have no experience in the US.

I think this paper is an important one but it needs a lot more work to reflect and represent  your data in the content of contemporary evidence and frameworks for (re)conceptualising healthcare policy and practice through a cultural lens.

Response to Reviewer Two

Major

  1. Since the event did take place in 2014 the authors have purposefully made sure to include literature that is more recent. The literature demonstrates there has been changes, particularly in Australia, like “Closing the Gap” (thank you for the suggestion). However, there are still gaps like the UN Sustainable Development Goals, which only mentions Indigenous Peoples in 4 of 169 metrics. (This is a topic included in this rewrite). See Lines 237-242 & 253-259.
  2. We have updated Figure One so that it is streamlined, more clear, and details in the text how it connects to health and healthcare.
  3. The methods section has been updated in more detail. It includes more information on recruitment, data, collection, and analysis including further citations of the different methods used. See Lines 173-197.

Minor

  1. The line now includes the dominant, power-holding, non-Indigenous, white population because the white population is the dominant population in the United States that has driven what health looks like in the US and is often used in comparison.
  2. Corrected the word error on Line 215, which is now Line 253

Reviewer 3 Report

This is an excellent research contribution- thank you.

One of the valuable aspects that I drew from this paper included the different definitions of health- resulting in the excellent Table 1 on 'Determinants of Health: Indigenous Knowledge and Western World Views'.

It may be of value to state the authors' Indigeneity if appropriate. I'm writing from Australia, and it is very familiar for Indigenous authors to include the name of their language group and traditional Country. It assists to ground the work from personal experience as well as academic expertise. More specifically, are all of the 5 Indigenous panel scholars also co-authors on this paper? And are the remaining co-authors also Indigenous?

There would be value in providing more details on the methods applied to compile these findings from the engagement of the scholars' panel.

It may be helpful to open with strengths-based statements before describing the inequities. This could be about resilience, survival, pride and identity of the First Peoples of the USA. It helps set the scene in terms of strength rather than deficit.

In Australia we tend not to use the term 'non-Indigenous, white populations' as there are many non-Indigenous residents, and the non-white population have a diversity of health status. Consider adjusting the use of 'white' to include additional non-Indigenous groups.

An additional reference of interest may be:

Personal View| Volume 6, ISSUE 2, e156-e163, February 01, 2022

The determinants of planetary health: an Indigenous consensus perspective

  • Nicole Redvers, ND
  • Yuria Celidwen, PhD
  • Clinton Schultz, PhD
  • Ojistoh Horn, MD
  • Cicilia Githaiga, MA
  • Melissa Vera, RN
  • Marlikka Perdrisat, BComm
  • Lynn Mad Plume, MPH
  • Daniel Kobei, MBA
  • Myrna Cunningham Kain, MD
  • Anne Poelina, PhD
  • Juan Nelson Rojas
  • Be'sha Blondin

Open AccessPublished:February, 2022DOI:https://doi.org/10.1016/S2542-5196(21)00354-5

Author Response

Reviewer Three

This is an excellent research contribution- thank you.

One of the valuable aspects that I drew from this paper included the different definitions of health- resulting in the excellent Table 1 on 'Determinants of Health: Indigenous Knowledge and Western World Views'.

It may be of value to state the authors' Indigeneity if appropriate. I'm writing from Australia, and it is very familiar for Indigenous authors to include the name of their language group and traditional Country. It assists to ground the work from personal experience as well as academic expertise. More specifically, are all of the 5 Indigenous panel scholars also co-authors on this paper? And are the remaining co-authors also Indigenous?

There would be value in providing more details on the methods applied to compile these findings from the engagement of the scholars' panel.

It may be helpful to open with strengths-based statements before describing the inequities. This could be about resilience, survival, pride and identity of the First Peoples of the USA. It helps set the scene in terms of strength rather than deficit.

In Australia we tend not to use the term 'non-Indigenous, white populations' as there are many non-Indigenous residents, and the non-white population have a diversity of health status. Consider adjusting the use of 'white' to include additional non-Indigenous groups.

Response to Reviewer Three

Please, see Lines 82-23 & 88 for an update about the Indigeneity of the authors. All 5 Indigenous panel scholars were also authors along with one of the remaining authors. This has been made more explicit on Lines 81-89.

The methods section (Lines 173-197) have been updated to include what methods were applied to compile the findings from the scholar’s panel.

The opening changed to include a more strengths-based statement. Thank you for that suggestion. It is something we do in our own work outside of this manuscript.

Sometimes we are comparing to the dominant structures, which were created by white settler populations so that is why non-Indigenous, white populations is used. In other places the term settler was used instead. We designate this description with the use of the word dominant.

Thank you for the reference. It was included in the paper Lines 324-330.

Round 2

Reviewer 2 Report

Thank you for the opportunity to again review this paper. I think this is important research. However, I remain concerned about the age of this data (7 years old) and that so much has happened in the intervening period with respect to Indigenous methodologies and decolonising theory. I appreciate that there is little that can be done about that other than to contextualise it to contemporary policy. 

A minor note: ref 126 is not correct, it should be https://www.closingthegap.gov.au/national-agreement/targets  

Author Response

Thank you for your comments. We understand the concerns about the age of the data and appreciate your understanding. We have updated the citation to refer to the partnership agreement itself.
